# KDGAN: Knowledge Distillation with Generative Adversarial Networks

**Xiaojie Wang**
University of Melbourne
xiaojiew94@gmail.com

**Rui Zhang**[*]
University of Melbourne
rui.zhang@unimelb.edu.au

**Yu Sun**
Twitter Inc.
ysun@twitter.com

**Jianzhong Qi**
University of Melbourne
jianzhong.qi@unimelb.edu.au

## Abstract

Knowledge distillation (KD) aims to train a lightweight classifier suitable to provide accurate inference with constrained resources in multi-label learning. Instead of directly consuming feature-label pairs, the classifier is trained by a teacher, i.e., a high-capacity model whose training may be resource-hungry. The accuracy of the classifier trained this way is usually suboptimal because it is difficult to learn the true data distribution from the teacher. An alternative method is to adversarially train the classifier against a discriminator in a two-player game akin to generative adversarial networks (GAN), which can ensure the classifier to learn the true data distribution at the equilibrium of this game. However, it may take excessively long time for such a two-player game to reach equilibrium due to high-variance gradient updates. To address these limitations, we propose a three-player game named KDGAN consisting of a classifier, a teacher, and a discriminator. The classifier and the teacher learn from each other via distillation losses and are adversarially trained against the discriminator via adversarial losses. By simultaneously optimizing the distillation and adversarial losses, the classifier will learn the true data distribution at the equilibrium. We approximate the discrete distribution learned by the classifier (or the teacher) with a concrete distribution. From the concrete distribution, we generate continuous samples to obtain low-variance gradient updates, which speed up the training. Extensive experiments using real datasets confirm the superiority of KDGAN in both accuracy and training speed.

## 1   Introduction

In machine learning, it is common that more resources such as input features [47] or computational resources [23], which we refer to as *privileged provision*, are available at the stage of training a model than those available at the stage of running the deployed model (i.e., the inference stage). Figure 1 shows an example application of image tag recommendation, where more input features (called *privileged information* [47]) are available at the training stage than those available at the inference stage. Specifically, the training stage has access to images as well as image titles and comments (textual information) as shown in Figure 1a, whereas the inference stage only has access to images themselves as shown in Figure 1b. After a smart phone user uploads an image and is about to provide tags for the image, it is inconvenient to type tags on the phone and thinking about tags for the image also takes time, so it is very useful to recommend tags based on the image as shown in Figure 1b. Another example application is unlocking mobile phones by face recognition. We usually deploy face recognition models on mobile phones so that legit users can unlock the phones without depending

---

[*]Corresponding author

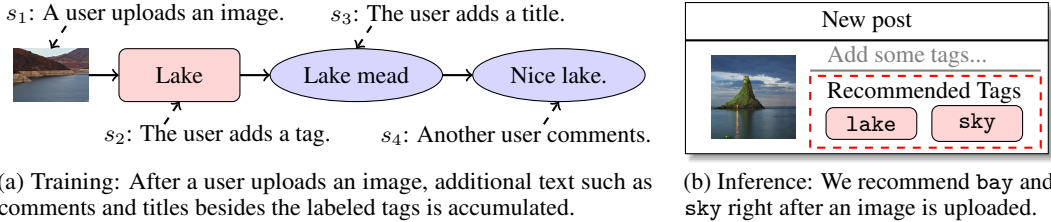

$s_1$: A user uploads an image.   $s_3$: The user adds a title.

Lake → Lake mead → Nice lake.

$s_2$: The user adds a tag.   $s_4$: Another user comments.

New post

Add some tags...

Recommended Tags

lake   sky

(a) Training: After a user uploads an image, additional text such as comments and titles besides the labeled tags is accumulated.

(b) Inference: We recommend `bay` and `sky` right after an image is uploaded.

Figure 1: Image tag recommendation where the additional text is only available for training.

on remote services or internet connections. The training stage may be done on a powerful server with significantly more computational resources than the inference stage, which is done on a mobile phone. Here, a key problem is how to use privileged provision, i.e., resources only accessible for training, to train a model with great inference performance [29].

Typical approaches to the problem are based on *knowledge distillation* (KD) [7, 9, 23]. As shown by the left half of Figure 2, KD consists of a *classifier* and a *teacher* [29]. To operate for resource-constrained inference, the classifier does not use privileged provision. On the other hand, the teacher uses privileged provision by, e.g., having a larger model capacity or taking more features as input. Once trained, the teacher outputs a distribution over labels called *soft labels* [29] for each training instance. Then, the teacher trains the classifier to predict the soft labels via a distillation loss such as the L2 loss on logits [7]. This training process is often called "distilling" the knowledge in the teacher into the classifier [23]. Since the teacher normally cannot perfectly model the true data distribution, it is difficult for the classifier to learn the true data distribution from the teacher.

Generative adversarial networks (GAN) provide an alternative way to learn the true data distribution. Inspired by Wang et al. [49], we first present a naive GAN (NaGAN) with two players. As shown by the right part of Figure 2, NaGAN consists of a classifier and a discriminator. The classifier serves as a generator that generates relevant labels given an instance while the discriminator aims to distinguish the true labels from the generated ones. The classifier learns from the discriminator to perfectly model the true data distribution at the equilibrium via adversarial losses. One limitation of NaGAN is that a large number of training instances and epochs is normally required to reach equilibrium [15], which restricts its applicability to domains where collecting labeled data is expensive. The slow training speed is because in such a two-player framework, the gradients from the discriminator to update the classifier often vanish or explode during the adversarial training [4]. It is challenging to train a classifier to learn the true data distribution with limited training instances and epochs.

To address this challenge, we propose a three-player framework named KDGAN to *distill knowledge with generative adversarial networks*. As shown in Figure 2, KDGAN consists of a *classifier*, a *teacher*, and a *discriminator*. In addition to the distillation loss in KD and the adversarial losses in NaGAN mentioned above, we define a distillation loss from the classifier to the teacher and an adversarial loss between the teacher and the discriminator. Specifically, the classifier and the teacher, serving as generators, aim to fool the discriminator by generating pseudo labels that resemble the true labels. Meanwhile, the classifier and the teacher try to reach an agreement on what pseudo labels to generate by distilling their knowledge into each other. By formulating the distillation and adversarial losses as a minimax game, we enable the classifier to learn the true data distribution at the equilibrium (see Section 3.2). Besides, the classifier receives gradients from the teacher via the distillation loss and the discriminator via the adversarial loss. The gradients from the teacher often have low variance, which reduces the variance of gradients and thus speeds up the adversarial training (see Section 3.3).

We further consider reducing the variance of the gradients from the discriminator to accelerate the training of KDGAN. The gradients from the discriminator may have large variance when obtained through the widely used policy gradient methods [49, 52]. It is non-trivial to obtain low-variance gradients from the discriminator because the classifier and the teacher generate discrete samples, which are not differentiable w.r.t. their parameters. We propose to relax the discrete distributions learned by the classifier and the teacher into concrete distributions [25, 31] with the Gumbel-Max trick [20, 30]. We use the concrete distributions for generating continuous samples to enable end-to-end differentiability and sufficient control over the variance of gradients. Given the continuous samples, we obtain low-variance gradients from the discriminator to accelerate the KDGAN training.

To summarize, our contributions are as follows:

- We propose a novel framework named KDGAN for multi-label learning, which trains a lightweight classifier suitable for resource-constrained inference using resources available only for training.
- We reduce the number of training epochs required to converge by decreasing the variance of gradients, which is achieved by the design of KDGAN and the Gumbel-Max trick.
- We conduct extensive experiments in two applications, image tag recommendation and deep model compression. The experiments validate the superiority of KDGAN over state-of-the-art methods.

## 2  Related Work

We briefly review studies on knowledge distillation (KD) and generative adversarial networks (GAN).

KD aims to transfer the knowledge in a powerful teacher to a lightweight classifier [9]. For example, Ba and Caruana [7] train a shallow classifier network to mimic a deep teacher network by matching logits via the L2 loss. Hinton et al. [23] generalize this work by training a classifier to predict soft labels provided by a teacher. Sau and Balasubramanian [39] further add random perturbations into soft labels to simulate learning from multiple teachers. Instead of using soft labels, Romero et al. [36] propose to use middle layers of a teacher to train a classifier. Unlike previous work on classification problems, Chen et al. [10] apply KD and hint learning to object detection problems. There also exists work that leverages KD to transfer knowledge between different domains [21], e.g., between high-quality and low-quality images [41]. Lopez-Paz et al. [29] unify KD with privileged information [35, 47, 48] as generalized distillation where a teacher is pretrained by taking as input privileged information. Compared to KD, the proposed KDGAN framework introduces a discriminator to guarantee that the classifier can learn the true data distribution at the equilibrium.

GAN is initially proposed to generate continuous data by training a generator and a discriminator adversarially in a minimax game [17]. GAN has only recently been introduced to generate discrete data [16, 54, 55] because discrete data makes it difficult to pass gradients from a discriminator backward to update a generator. For example, sequence GAN (SeqGAN) [52] models the process of token sequence generation as a stochastic policy and adopts Monte Carlo search to update a generator. Different from these GANs with two players, Li et al. propose a GAN with three players called Triple-GAN [13]. Our KDGAN also consists of three players including two generators and a discriminator, but differs from Triple-GAN in that: (1) Both generators in KDGAN learn a conditional distribution over labels given features. However, the generators in Triple-GAN learn a conditional distribution over labels given features and a conditional distribution over features given labels, respectively. (2) The samples from both generators in KDGAN are all discrete data while the samples from the generators in Triple-GAN include both discrete and continuous data. These differences lead to different objective functions and training techniques, e.g., KDGAN can use the Gumbel-Max trick [20, 30] to generate samples from both generators while Triple-GAN cannot do this. There is also a rich body of studies on improving the training of GAN [5, 33, 56] such as feature matching [38], which are orthogonal to our work and can be used to improve the training of KDGAN.

We explore the idea of integrating KD and GAN. A similar idea has been studied in [51] where a discriminator is introduced to train a classifier. This previous study [51] differs from ours in that their discriminator trains the classifier to learn the data distribution produced by the teacher, while our discriminator trains the classifier to learn the true data distribution.

We apply the proposed KDGAN to address the problem of deep model compression and image tag recommendation. We can also apply KDGAN to address the other problems where privileged provision is available [44]. For example, we can consider contextual signals in the intent tracking problem [42, 43] or user reviews in the movie recommendation problem [50] as privileged provision.

## 3  Methods

We study the problem of training a lightweight classifier from a teacher that is trained with privileged provision (denoted by $\varrho$) to satisfy stringent inference requirements. The inference requirements may include (1) running in real time with limited computational resources, where privileged provision is computational resources [23]; (2) lacking a certain type of input features, where privileged provision is privileged information [47]. Following existing work [29], we use multi-label learning problems [12, 18, 53] as the target application scenarios of our methods for illustration purpose.

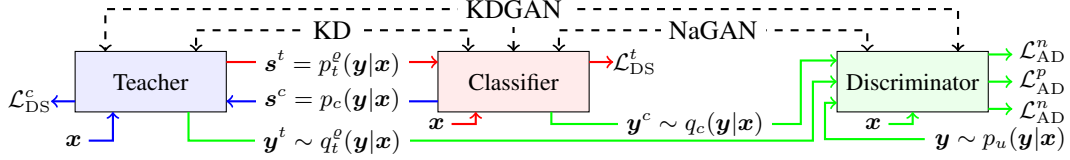

Figure 2: Comparison among KD, NaGAN, and KDGAN. The classifier ($C$) and the teacher ($T$) learn discrete categorical distributions $p_c(\boldsymbol{y}|\boldsymbol{x})$ and $p_t^{\varrho}(\boldsymbol{y}|\boldsymbol{x})$; $\boldsymbol{y}$ is a true label generated from the true data distribution $p_u(\boldsymbol{y}|\boldsymbol{x})$; $\boldsymbol{y}^c$ and $\boldsymbol{y}^t$ are continuous samples generated from concrete distributions $q_c(\boldsymbol{y}|\boldsymbol{x})$ and $q_t^{\varrho}(\boldsymbol{y}|\boldsymbol{x})$; $\boldsymbol{s}^c$ and $\boldsymbol{s}^t$ are soft labels produced by $C$ and $T$; $\mathcal{L}_{\text{DS}}^c$ and $\mathcal{L}_{\text{DS}}^t$ are distillation losses for $C$ and $T$; $\mathcal{L}_{\text{AD}}^p$ and $\mathcal{L}_{\text{AD}}^n$ are adversarial losses for positive and negative feature-label pairs.

Since privileged provision is only available at the training stage, the goal of the problem is to train a lightweight classifier that does not use privileged provision for effective inference.

To achieve this goal, we start with NaGAN, a naive adaptation of the two-player framework proposed by Wang et al. in information retrieval (Section 3.1). Similar to other two-player frameworks [49], the naive adaptation requires a large number of training instances and epochs [15], which is difficult to satisfy in practice [4]. To address the limitation, we propose a three-player framework named KDGAN that can speed up the training while preserving the equilibrium (Sections 3.2 and 3.3).

## 3.1 NaGAN Formulation

We begin with NaGAN that combines a classifier $C$ with a discriminator $D$ in a minimax game. Since $D$ is not meant for inference, it can leverage privileged provision. For example, $D$ may have a larger model capacity than $C$ or take as input more features than those available to $C$. In NaGAN, $C$ generates pseudo labels $\boldsymbol{y}$ given features $\boldsymbol{x}$ following a categorical distribution $p_c(\boldsymbol{y}|\boldsymbol{x})$, while $D$ computes the probability $p_d^{\varrho}(\boldsymbol{x}, \boldsymbol{y})$ of a label $\boldsymbol{y}$ being from the true data distribution $p_u(\boldsymbol{y}|\boldsymbol{x})$ given features $\boldsymbol{x}$. With a slight abuse of notation, we also use $\boldsymbol{x}$ to refer to features including privileged information when the context is clear. Following the value function of IRGAN [49], we define the value function $V(c, d)$ for the minimax game in NaGAN as

$$\min_c \max_d V(c, d) = \mathbb{E}_{\boldsymbol{y} \sim p_u}[\log p_d^{\varrho}(\boldsymbol{x}, \boldsymbol{y})] + \mathbb{E}_{\boldsymbol{y} \sim p_c}[\log(1 - p_d^{\varrho}(\boldsymbol{x}, \boldsymbol{y}))]. \tag{1}$$

Let $h(\boldsymbol{x}, \boldsymbol{y})$ and $g(\boldsymbol{x}, \boldsymbol{y})$ be the scoring functions for $C$ and $D$. We define $p_c(\boldsymbol{y}|\boldsymbol{x})$ and $p_d^{\varrho}(\boldsymbol{x}, \boldsymbol{y})$ as

$$p_c(\boldsymbol{y}|\boldsymbol{x}) = \text{softmax}(h(\boldsymbol{x}, \boldsymbol{y})) \quad \text{and} \quad p_d^{\varrho}(\boldsymbol{x}, \boldsymbol{y}) = \text{sigmoid}(g(\boldsymbol{x}, \boldsymbol{y})). \tag{2}$$

The scoring functions can be implemented in various ways, e.g., $h(\boldsymbol{x}, \boldsymbol{y})$ can be a multilayer perceptron [27]. We will detail the scoring functions for specific applications in Section 4. Such a two-player framework is trained by updating $C$ and $D$ alternatively [49]. The training will proceed until the equilibrium is reached, where $C$ learns the true data distribution. At that point, $D$ can do no better than random guesses at deciding whether a given label is generated by $C$ or not [6].

Our key observation is that the advantages and the disadvantages of KD and NaGAN are complementary: (1) KD usually requires a small number of training instances and epochs but cannot ensure the equilibrium where $p_c(\boldsymbol{y}|\boldsymbol{x}) = p_u(\boldsymbol{y}|\boldsymbol{x})$. (2) NaGAN ensures the equilibrium where $p_c(\boldsymbol{y}|\boldsymbol{x}) = p_u(\boldsymbol{y}|\boldsymbol{x})$ [49] but normally requires a large number of training instances and epochs. We aim to retain the advantages and avoid the disadvantages of both methods in a single framework.

## 3.2 KDGAN Formulation

We formulate KDGAN as a minimax game with a classifier $C$, a teacher $T$, and a discriminator $D$. Similar to the classifier $C$, the teacher $T$ generates pseudo labels based on a categorical distribution $p_t^{\varrho}(\boldsymbol{y}|\boldsymbol{x}) = \text{softmax}(f(\boldsymbol{x}, \boldsymbol{y}))$ where $f(\boldsymbol{x}, \boldsymbol{y})$ is also a scoring function. Both $T$ and $D$ use privileged provision, e.g., by having a large model capacity or taking privileged information as input. In KDGAN, $D$ aims to maximize the probability of correctly distinguishing the true and pseudo labels, whereas $C$ and $T$ aim to minimize the probability that $D$ rejects their generated pseudo labels. Meanwhile, $C$ learns from $T$ by mimicking the learned distribution of $T$. To build a general framework, we also enable $T$ to learn from $C$ because, in reality, a teacher's ability can also be enhanced by interacting with students (see Figure 6 in Appendix D for empirical evidence that $T$ benefits from learning from

**Algorithm 1:** Minibatch stochastic gradient descent training of KDGAN.

---

1 Pretrain a classifier $C$, a teacher $T$, and a discriminator $D$ with the training data $\{(\boldsymbol{x}_1, \boldsymbol{y}_1), ..., (\boldsymbol{x}_n, \boldsymbol{y}_n)\}$.
2 **for** *the number of training epochs* **do**
3     **for** *the number of training steps for the discriminator* **do**
4         Sample labels $\{\boldsymbol{y}_1, ..., \boldsymbol{y}_k\}$, $\{\boldsymbol{y}_1^c, ..., \boldsymbol{y}_k^c\}$, and $\{\boldsymbol{y}_1^t, ..., \boldsymbol{y}_k^t\}$ from $p_u(\boldsymbol{y}|\boldsymbol{x})$, $q_c(\boldsymbol{y}|\boldsymbol{x})$, and $q_t^\varrho(\boldsymbol{y}|\boldsymbol{x})$.
5         Update $D$ by ascending along its gradients
6         $\frac{1}{k}\sum_{i=1}^{k}\left(\nabla_d \log p_d^\varrho(\boldsymbol{x}, \boldsymbol{y}_i) + \alpha\nabla_d \log(1 - p_d^\varrho(\boldsymbol{x}, \boldsymbol{z}_i^c)) + (1-\alpha)\nabla_d \log(1 - p_d^\varrho(\boldsymbol{x}, \boldsymbol{z}_i^t))\right)$.
7     **for** *the number of training steps for the teacher* **do**
8         Sample labels $\{\boldsymbol{y}_1^t, ..., \boldsymbol{y}_k^t\}$ from $q_t^\varrho(\boldsymbol{y}|\boldsymbol{x})$ and update the teacher by descending along its gradients
9         $\frac{1}{k}\sum_{i=1}^{k}(1-\alpha)\nabla_t \log q_t^\varrho(\boldsymbol{y}_i^t|\boldsymbol{x})\log(1 - p_d^\varrho(\boldsymbol{x}, \boldsymbol{z}_i^t)) + \gamma\nabla_t \mathcal{L}_{\mathrm{DS}}^t(p_t^\varrho(\boldsymbol{y}|\boldsymbol{x}), p_c(\boldsymbol{y}|\boldsymbol{x}))$.
10     **for** *the number of training steps for the classifier* **do**
11         Sample labels $\{\boldsymbol{y}_1^c, ..., \boldsymbol{y}_k^c\}$ from $q_c(\boldsymbol{y}|\boldsymbol{x})$ and update $C$ by descending along its gradients
12         $\frac{1}{k}\sum_{i=1}^{k}\alpha\nabla_c \log q_c(\boldsymbol{y}_i^c|\boldsymbol{x})\log(1 - p_d^\varrho(\boldsymbol{x}, \boldsymbol{z}_i^c)) + \beta\nabla_c \mathcal{L}_{\mathrm{DS}}^c(p_c(\boldsymbol{y}|\boldsymbol{x}), p_t^\varrho(\boldsymbol{y}|\boldsymbol{x}))$.

---

$C$). Such a mutual learning helps $C$ and $T$ reduce their probability of generating different pseudo labels. Formally, we define the value function $U(c, t, d)$ for the minimax game in KDGAN as

$$\min_{c,t}\max_{d} U(c, t, d) = \mathbb{E}_{\boldsymbol{y}\sim p_u}[\log p_d^\varrho(\boldsymbol{x}, \boldsymbol{y})] + \alpha\mathbb{E}_{\boldsymbol{y}\sim p_c}[\log(1 - p_d^\varrho(\boldsymbol{x}, \boldsymbol{y}))]$$
$$+ (1-\alpha)\mathbb{E}_{\boldsymbol{y}\sim p_t^\varrho}[\log(1 - p_d^\varrho(\boldsymbol{x}, \boldsymbol{y}))] + \beta\mathcal{L}_{\mathrm{DS}}^c(p_c(\boldsymbol{y}|\boldsymbol{x}), p_t^\varrho(\boldsymbol{y}|\boldsymbol{x})) + \gamma\mathcal{L}_{\mathrm{DS}}^t(p_t^\varrho(\boldsymbol{y}|\boldsymbol{x}), p_c(\boldsymbol{y}|\boldsymbol{x})), \quad (3)$$

where $\alpha \in (0, 1)$, $\beta \in (0, +\infty)$, and $\gamma \in (0, +\infty)$ are hyperparameters. We collectively refer to the expectation terms as the *adversarial losses* and refer to $\mathcal{L}_{\mathrm{DS}}^c$ and $\mathcal{L}_{\mathrm{DS}}^t$ as the *distillation losses*. The distillation losses can be defined in several ways [39], e.g., the L2 loss [7] or Kullback–Leibler divergence [23]. Note that $\mathcal{L}_{\mathrm{DS}}^c$ and $\mathcal{L}_{\mathrm{DS}}^t$ are used to train the classifier and the teacher, respectively.

**Theoretical Analysis**. We show that the classifier perfectly learns the true data distribution at the equilibrium of KDGAN. To see this, let $p_\alpha^\varrho(\boldsymbol{y}|\boldsymbol{x}) = \alpha p_c(\boldsymbol{y}|\boldsymbol{x}) + (1-\alpha)p_t^\varrho(\boldsymbol{y}|\boldsymbol{x})$. It can be shown that the adversarial losses w.r.t. $p_c(\boldsymbol{y}|\boldsymbol{x})$ and $p_t^\varrho(\boldsymbol{y}|\boldsymbol{x})$ are equal to an adversarial loss w.r.t. $p_\alpha^\varrho(\boldsymbol{y}|\boldsymbol{x})$:

$$\alpha\mathbb{E}_{\boldsymbol{y}\sim p_c}[\log(1 - p_d^\varrho(\boldsymbol{x}, \boldsymbol{y}))] + (1-\alpha)\mathbb{E}_{\boldsymbol{y}\sim p_t^\varrho}[\log(1 - p_d^\varrho(\boldsymbol{x}, \boldsymbol{y}))]$$
$$= \alpha\sum_{\boldsymbol{y}} p_c(\boldsymbol{y}|\boldsymbol{x})\log(1 - p_d^\varrho(\boldsymbol{x}, \boldsymbol{y})) + (1-\alpha)\sum_{\boldsymbol{y}} p_t^\varrho(\boldsymbol{y}|\boldsymbol{x})\log(1 - p_d^\varrho(\boldsymbol{x}, \boldsymbol{y}))$$
$$= \sum_{\boldsymbol{y}}\left(\alpha p_c(\boldsymbol{y}|\boldsymbol{x}) + (1-\alpha)p_t^\varrho(\boldsymbol{y}|\boldsymbol{x})\right)\log(1 - p_d^\varrho(\boldsymbol{x}, \boldsymbol{y})) \quad (4)$$
$$= \mathbb{E}_{\boldsymbol{y}\sim p_\alpha^\varrho}[\log(1 - p_d^\varrho(\boldsymbol{x}, \boldsymbol{y}))].$$

Therefore, let $\mathcal{L}_{\mathrm{MD}} = \beta\mathcal{L}_{\mathrm{DS}}^c(p_c(\boldsymbol{y}|\boldsymbol{x}), p_t^\varrho(\boldsymbol{y}|\boldsymbol{x})) + \gamma\mathcal{L}_{\mathrm{DS}}^t(p_t^\varrho(\boldsymbol{y}|\boldsymbol{x}), p_c(\boldsymbol{y}|\boldsymbol{x}))$ and $\mathcal{L}_{\mathrm{JS}}$ be the Jensen-Shannon divergence, the value function $U(c, t, d)$ of the minimax game can be rewritten as

$$\min_{\alpha}\max_{d}\mathbb{E}_{\boldsymbol{y}\sim p_u}[\log p_d^\varrho(\boldsymbol{x}, \boldsymbol{y})] + \mathbb{E}_{\boldsymbol{y}\sim p_\alpha^\varrho}[\log(1 - p_d^\varrho(\boldsymbol{x}, \boldsymbol{y}))] + \mathcal{L}_{\mathrm{MD}} \quad (5)$$
$$= \min_{\alpha} 2\mathcal{L}_{\mathrm{JS}}(p_u(\boldsymbol{y}|\boldsymbol{x})||p_\alpha^\varrho(\boldsymbol{y}|\boldsymbol{x})) + \beta\mathcal{L}_{\mathrm{DS}}^c(p_c(\boldsymbol{y}|\boldsymbol{x}), p_t^\varrho(\boldsymbol{y}|\boldsymbol{x})) + \gamma\mathcal{L}_{\mathrm{DS}}^t(p_t^\varrho(\boldsymbol{y}|\boldsymbol{x}), p_c(\boldsymbol{y}|\boldsymbol{x})) - \log(4).$$

Here, $\mathcal{L}_{\mathrm{JS}}$ reaches the minimum if and only if $p_\alpha^\varrho(\boldsymbol{y}|\boldsymbol{x}) = p_u(\boldsymbol{y}|\boldsymbol{x})$ and $\mathcal{L}_{\mathrm{DS}}^c$ (or $\mathcal{L}_{\mathrm{DS}}^t$) reaches the minimum if and only if $p_c(\boldsymbol{y}|\boldsymbol{x}) = p_t^\varrho(\boldsymbol{y}|\boldsymbol{x})$. Hence, the KDGAN equilibrium is reached if and only if $p_c(\boldsymbol{y}|\boldsymbol{x}) = p_t^\varrho(\boldsymbol{y}|\boldsymbol{x}) = p_u(\boldsymbol{y}|\boldsymbol{x})$ where the classifier learns the true data distribution. We summarize the above discussions in Lemma 4.1 (the necessary and sufficient conditions of maximizing the value function) and Theorem 4.2 (achieving the equilibrium), respectively (see Appendix A for proofs).

**Lemma 4.1.** *For any fixed classifier and teacher, the value function $U(c, t, d)$ is maximized if and only if the distribution of the discriminator is given by $p_d^\varrho(\boldsymbol{x}, \boldsymbol{y}) = {p_u(\boldsymbol{y}|\boldsymbol{x})}/{(p_u(\boldsymbol{y}|\boldsymbol{x}) + p_\alpha^\varrho(\boldsymbol{y}|\boldsymbol{x}))}$.*

**Theorem 4.2.** *The equilibrium of the minimax game $\min_{c,t}\max_d U(c, t, d)$ is achieved if and only if $p_c(\boldsymbol{y}|\boldsymbol{x}) = p_t^\varrho(\boldsymbol{y}|\boldsymbol{x}) = p_u(\boldsymbol{y}|\boldsymbol{x})$. At that point, $U(c, t, d)$ reaches the value $-\log(4)$.*

### 3.3 KDGAN Training

In this section, we detail techniques for accelerating the training speed of KDGAN via reducing the number of training epochs needed. As discussed in earlier studies [8, 46], the training speed is closely related to the variance of gradients. Comparing with NaGAN, the KDGAN framework by design can reduce the variance of gradients. This is because the high variance of a random variable can

be reduced by a low-variance random variable (detailed in Lemma 4.3) and as we will discuss, $T$ provides gradients of lower variance than $D$ does. To reduce the variance of gradients from $D$ and attain sufficient control over the variance, we further propose to obtain gradients from a continuous space by relaxing the discrete samples, i.e., pseudo labels, propagated between the classifier (or the teacher) and the discriminator into continuous samples with a reparameterization trick [25, 31].

First, we show how KDGAN reduces the variance of gradients. As discussed above, $C$ only receives gradients $\nabla_c V$ from $D$ in NaGAN while it receives gradients $\nabla_c U$ from both $D$ and $T$ in KDGAN:

$$\nabla_c V = \nabla_c \mathcal{L}_{\mathrm{AD}}^n, \quad \nabla_c U = \lambda \nabla_c \mathcal{L}_{\mathrm{AD}}^n + (1 - \lambda) \nabla_c \mathcal{L}_{\mathrm{DS}}^c, \qquad (6)$$

where $\lambda \in (0, 1)$, $\nabla_c \mathcal{L}_{\mathrm{AD}}^n$ and $\nabla_c \mathcal{L}_{\mathrm{DS}}^c$ are gradients from $D$ and $T$, respectively. Consistent with the findings in existing work [23, 39], we also observe that $\nabla_c \mathcal{L}_{\mathrm{DS}}^c$ usually has a lower variance than $\nabla_c \mathcal{L}_{\mathrm{AD}}^n$ (see Figure 7 in Appendix D for empirical evidence that the variance of $\nabla_c \mathcal{L}_{\mathrm{DS}}^c$ is smaller than that of $\nabla_c \mathcal{L}_{\mathrm{AD}}^n$ during the training process). Hence, it can be easily shown that the gradients w.r.t. $C$ in KDGAN have a lower variance than that in NaGAN (refer to Lemma 4.3):

$$\mathrm{Var}(\nabla_c \mathcal{L}_{\mathrm{DS}}^c) \leq \mathrm{Var}(\nabla_c \mathcal{L}_{\mathrm{AD}}^n) \Rightarrow \mathrm{Var}(\nabla_c U) \leq \mathrm{Var}(\nabla_c V). \qquad (7)$$

Next, we further reduce the variance of gradients with a reparameterization trick, in particular, the Gumbel-Max trick [20, 30]. The essence of the Gumbel-Max trick is to reparameterize generating discrete samples into a differentiable function of its parameters and an additional random variable of a Gumbel distribution. To perform the Gumbel-Max trick on generating discrete samples from the categorical distribution $p_c(\boldsymbol{y}|\boldsymbol{x})$, a concrete distribution [25, 31] can be used. We use a concrete distribution $q_c(\boldsymbol{y}|\boldsymbol{x})$ to generate continuous samples and use the continuous samples to compute the gradients $\nabla_c \mathcal{L}_{\mathrm{AD}}^n$ of the adversarial loss w.r.t. the classifier as

$$\nabla_c \mathcal{L}_{\mathrm{AD}}^n = \nabla_c \mathbb{E}_{\boldsymbol{y} \sim p_c}[\log(1 - p_d^o(\boldsymbol{x}, \boldsymbol{y}))] = \mathbb{E}_{\boldsymbol{y} \sim q_c}[\nabla_c \log q_c(\boldsymbol{y}|\boldsymbol{x}) \log(1 - p_d^o(\boldsymbol{x}, \boldsymbol{z}))]. \qquad (8)$$

Here, $\boldsymbol{z} = \mathrm{onehot}(\arg\max \boldsymbol{y})$ is a discrete pseudo label where $\boldsymbol{y} \sim q_c(\boldsymbol{y}|\boldsymbol{x})$. We define $q_c(\boldsymbol{y}|\boldsymbol{x})$ as

$$q_c(\boldsymbol{y}|\boldsymbol{x}) = \mathrm{softmax}\left(\frac{\log p_c(\boldsymbol{y}|\boldsymbol{x}) + \boldsymbol{g}}{\tau}\right), \quad \boldsymbol{g} \sim \mathrm{Gumbel}(0, 1). \qquad (9)$$

Here, $\tau \in (0, +\infty)$ is a temperature parameter and $\mathrm{Gumbel}(0, 1)$ is the Gumbel distribution[2] [31]. We leverage the temperature parameter $\tau$ to control the variance of gradients over the training. With a high temperature, the samples from the concrete distribution are smooth, which give low-variance gradient estimates. Note that a disadvantage of the concrete distribution is that with a high temperature, it becomes a less accurate approximation to the original categorical distribution, which causes biased gradient estimates. We will discuss how to tune the temperature parameter in Section 4.

In addition to improving the training of $C$, we also apply the same techniques to improve the training of $T$. We update $D$ with the back-propagation algorithm [37] (detailed in Appendix B). The overall logic of the KDGAN training is summarized in Algorithm 1. The three players can be first pretrained separately and then trained alternatively via minibatch stochastic gradient descent.

## 4 Experiments

The proposed KDGAN framework can be applied to a wide range of multi-label learning tasks where privileged provision is available. To show the applicability of KDGAN, we conduct experiments with the tasks of deep model compression (Section 4.1) and image tag recommendation (Section 4.2). Note that privileged provision is referred to as computational resources in deep model compression and privileged information in image tag recommendation, respectively.

We implement KDGAN based on Tensorflow [1] and here we briefly describe our experimental setup[3]. We use two formulations of the distillation losses including the L2 loss [7] and the Kullback–Leibler divergence [23]. The two formulations exhibit comparable results and the results presented are based on the L2 loss [7]. Since both $T$ and $D$ can use privileged provision, we implement their scoring functions $f(\boldsymbol{x}, \boldsymbol{y})$ and $g(\boldsymbol{x}, \boldsymbol{y})$ using the same function $s(\boldsymbol{x}, \boldsymbol{y})$ but with different sets of parameters. We search for the optimal values for the hyperparameters $\alpha$ in $[0.0, 1.0]$, $\beta$ in $[0.001, 1000]$, and $\gamma$ in $[0.0001, 100]$ based on validation performance. We find that a reasonable annealing schedule for the temperature parameter $\tau$ is to start with a large value (1.0) and exponentially decay it to a small value (0.1). We leave the exploration of the optimal schedule for future work.

Table 1: Average accuracy over 10 runs in model compression ($n$ is the number of training instances).

| Method | MNIST | | | CIFAR-10 | | |
|---|---|---|---|---|---|---|
| | $n = 100$ | $n = 1,000$ | $n = 10,000$ | $n = 500$ | $n = 5,000$ | $n = 50,000$ |
| CODIS | $74.02 \pm 0.13$ | $95.77 \pm 0.10$ | $98.89 \pm 0.08$ | $54.17 \pm 0.20$ | $77.82 \pm 0.14$ | $85.12 \pm 0.11$ |
| DISTN | $68.34 \pm 0.06$ | $93.97 \pm 0.08$ | $98.79 \pm 0.07$ | $50.92 \pm 0.18$ | $76.59 \pm 0.15$ | $83.32 \pm 0.08$ |
| NOISY | $66.53 \pm 0.18$ | $93.45 \pm 0.11$ | $98.58 \pm 0.11$ | $50.18 \pm 0.28$ | $75.42 \pm 0.19$ | $82.99 \pm 0.12$ |
| MIMIC | $67.35 \pm 0.15$ | $93.78 \pm 0.13$ | $98.65 \pm 0.05$ | $51.74 \pm 0.23$ | $75.66 \pm 0.17$ | $84.33 \pm 0.10$ |
| NaGAN | $64.90 \pm 0.31$ | $93.60 \pm 0.22$ | $98.95 \pm 0.19$ | $46.29 \pm 0.32$ | $76.11 \pm 0.24$ | $85.34 \pm 0.27$ |
| KDGAN | $\mathbf{77.95} \pm 0.05$ | $\mathbf{96.42} \pm 0.05$ | $\mathbf{99.25} \pm 0.02$ | $\mathbf{57.56} \pm 0.13$ | $\mathbf{79.36} \pm 0.04$ | $\mathbf{86.50} \pm 0.04$ |

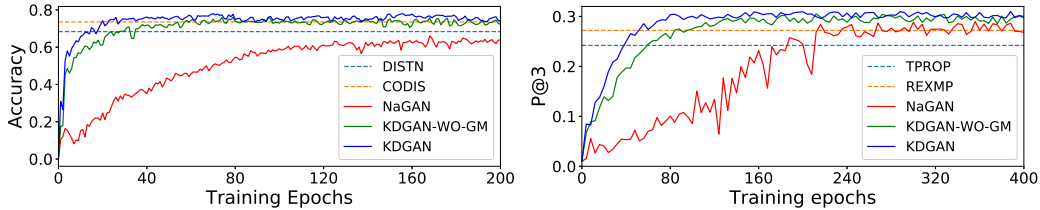

(a) Deep model compression over MNIST.　　　(b) Image tag recommendation on YFCC100M.

Figure 3: Training curves of the classifier in the proposed NaGAN and KDGAN.

## 4.1 Deep Model Compression

Deep model compression aims to reduce the storage and runtime complexity of deep models and to improve the deployability of such models on portable devices such as smart phones. Extensive computational resources available for training are considered privileged provision in this task.

**Dataset and Setup**. We use the widely adopted MNIST [27] and CIFAR-10 [26] datasets. The MNIST dataset has 60,000 grayscale images (50,000 for training and 10,000 for testing) with 10 different label classes. Following an earlier work [39], we do not preprocess the images on MNIST. The CIFAR-10 dataset has 60,000 colored images (50,000 for training and 10,000 for testing) with 10 different label classes. We preprocess the images by subtracting per-pixel mean, and we augment the training data by mirrored images. We vary the number of training instances in $[100, 10000]$ on MNIST and in $[500, 50000]$ on CIFAR-10. The scoring functions $h(\boldsymbol{x}, \boldsymbol{y})$ and $s(\boldsymbol{x}, \boldsymbol{y})$ are implemented as an MLP (1.2M parameters) and a LeNet (3.1M parameters) on MNIST; while $h(\boldsymbol{x}, \boldsymbol{y})$ and $s(\boldsymbol{x}, \boldsymbol{y})$ are implemented as a LeNet (0.5M parameters) and a ResNet (1.7M parameters) on CIFAR-10 (detailed in Appendix C). We evaluate various methods over 10 runs with different initialization of $C$ and report the mean accuracy and the standard deviation. Since the focus of this paper is to achieve a better accuracy for a given architecture of the classifier, we defer the discussion on the classifier's ratio of compression and loss of accuracy w.r.t. the teacher to Table 3 in Appendix D.

**Results and Discussions**. First, we compare the proposed NaGAN and KDGAN with KD-based methods including MIMIC [7], DISTN [23], NOISY [39], and CODIS [2]. The results obtained by varying the number of training images on MNIST and CIFAR-10 are summarized in Table 1. On both datasets, KDGAN consistently outperforms the KD-based methods by a large margin. For example, KDGAN achieves as much as 5.31% performance gain with 100 training images on MNIST. We further compare NaGAN with the KD-based methods. We observe that NaGAN performs better when a large amount of training data are available (e.g., 50,000 training images on CIFAR-10) while KD-based methods perform better when a small number of training images are available (e.g., 500 training images on CIFAR-10). This is consistent with our analysis in Section 3.1 that NaGAN can learn the true data distribution better, although this requires a large amount of training data.

Then, we compare NaGAN with KDGAN. As shown in Table 1, KDGAN achieves a larger performance gain over NaGAN with fewer training instances. This indicates that KDGAN requires a smaller number of training instances than NaGAN does to reach the same level of accuracy. This can be explained by that KDGAN introduces $T$ to provide soft labels for training $C$. The soft labels generally have high entropy and reveal much useful information about each training instance. Hence, the soft labels impose much more constraint on the parameters of $C$ than the true labels, which can reduce the number of training instances required to train $C$. We further investigate the training speed

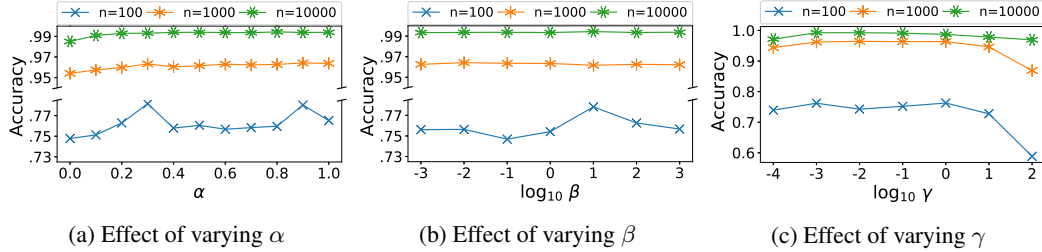

(a) Effect of varying $\alpha$      (b) Effect of varying $\beta$      (c) Effect of varying $\gamma$

Figure 4: Effects of hyperparameters in KDGAN on MNIST for deep model compression.

of NaGAN and KDGAN by the number of training epochs. Typical learning curves of $C$ in NaGAN and KDGAN are shown in Figure 3a. Due to the page limit, we only show the results using 100 training images on MNIST. We find that KDGAN converges to a better accuracy with a smaller number of training epochs (about 25 epochs) than NaGAN (about 135 epochs). After convergence, the training curve in KDGAN is more stable than that in NaGAN. Moreover, we investigate the benefit provided by the Gumbel-Max trick for the KDGAN training. We perform the KDGAN training without using the Gumbel-Max trick (referred to as KDGAN-WO-GM) and also plot the accuracy against training epochs in Figure 3a. By comparing KDGAN with KDGAN-WO-GM, we can see that the Gumbel-Max trick speeds up the training process by around 45% in terms of training epochs. The Gumbel-Max trick also helps improve the accuracy from 0.7605 to 0.7795 (by around 2.5%). One possible reason is that the Gumbel-Max trick effectively reduces the gradient variance from the discriminator as discussed in Section 3.3. This is also observed in our experiments, e.g., by comparing the gradient variance from the adversarial loss not using the Gumbel-Max trick in Figure 7a with the one using the Gumbel-Max trick in Figure 7b (see Appendix D for details).

Next, we study the reasons for the higher accuracy of KDGAN. We present how the accuracy of KDGAN varies against the hyperparameters on the MNIST dataset in Figure 4 (Note the logarithmic scale of the $x$-axis in Figures 4b and 4c). We find that $\alpha$ and $\beta$ have a relatively small effect on the accuracy, which suggests that KDGAN is a robust framework. Besides, if we set $\beta$ to a small value (0.0001), we get more than 2% accuracy drop when KDGAN is trained with 100 training instances. This shows that $T$ is important in training $C$ when the number of training instances is small. We further find that a large value of $\gamma$ causes the accuracy to deteriorate rapidly. This is because the soft labels provided by $C$ are usually noisy. Emphasizing on training $T$ to predict the noisy labels decreases the accuracy of $T$, which in turn decreases the accuracy of $C$. We obtain similar results for the effects of the hyperparameters on the CIFAR-10 dataset.

## 4.2 Image Tag Recommendation

Image tag recommendation aims to recommend relevant tags (i.e., labels) after a user uploads an image to image-hosting websites such as Flickr[4]. As discussed before, we aim to recommend relevant tags right after a user uploads an image. This way, the user can just select from the recommended tags instead of inputting tags. Users may continue to add additional text for an uploaded image such as image titles and descriptions. We only use such additional text at the training stage as privileged information used by the teacher and the discriminator only. At the inference stage, our trained model (i.e., the classifier) only takes an image as input to make tag recommendations.

**Dataset and Setup**. We use the Yahoo Flickr Creative Commons 100 Million (YFCC100M) dataset[5] in the experiments [45]. To simulate the case where additional text about images is available for training, we randomly sample 20,000 images with titles or descriptions for training and another 2,000 images for testing. We create a dataset of images labeled with the 200 most popular tags and another dataset of images labeled with 200 randomly sampled tags. Following an earlier study [3], we use a VGGNet [40] pretrained on ImageNet [14] to extract image features and a LSTM [24] with pretrained word embeddings [34] to learn text features. We implement $h(\boldsymbol{x}, \boldsymbol{y})$ as an MLP with image features as input and implement $s(\boldsymbol{x}, \boldsymbol{y})$ as an MLP with the element-wise product of image and text features as input (detailed in Appendix C). We use precision (P@N), F-score (F@N), mean average precision (MAP), and mean reciprocal ranking (MRR) to evaluate performance.

Table 2: Performance of various methods on the YFCC100M dataset in tag recommendation.

| Method | Most Popular Tags | | | | | | Randomly Sampled Tags | | | | | |
|--------|------|------|------|------|------|------|------|------|------|------|------|------|
| | P@3 | P@5 | F@3 | F@5 | MAP | MRR | P@3 | P@5 | F@3 | F@5 | MAP | MRR |
| KNN | .2320 | .1680 | .2339 | .1633 | .5755 | .5852 | .1623 | .1198 | .1575 | .1088 | .3970 | .4092 |
| TPROP | .2420 | .1636 | .2811 | .1949 | .6177 | .6270 | .1883 | .1372 | .1810 | .1252 | .4512 | .4636 |
| TFEAT | .2560 | .1752 | .2871 | .1999 | .6417 | .6503 | .2002 | .1420 | .2195 | .1495 | .5149 | .5309 |
| REXMP | .2720 | .1800 | .3324 | .2295 | .7015 | .7122 | .2228 | .1378 | .2427 | .1669 | .5205 | .5331 |
| NaGAN | .2892 | .1880 | .3516 | .2352 | .7432 | .7555 | .2415 | .1495 | .2693 | .1867 | .5791 | .5911 |
| KDGAN | **.3047** | **.1968** | **.3678** | **.2526** | **.7787** | **.7905** | **.2572** | **.1666** | **.2946** | **.2009** | **.6302** | **.6452** |

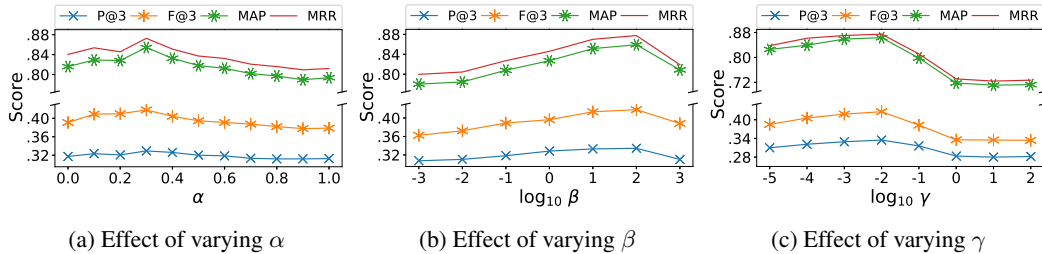

| (a) Effect of varying $\alpha$ | (b) Effect of varying $\beta$ | (c) Effect of varying $\gamma$ |
|---|---|---|

Figure 5: Effects of hyperparameters in KDGAN on YFCC100M for image tag recommendation.

**Results and Discussions**. First, we compare $C$ in KDGAN with KNN [32], TPROP [19], TFEAT [11], and REXMP [28]. The overall results are presented in Table 2. We find that KDGAN achieves significant improvements over the other methods across all the measures. Although KDGAN does not explicitly model the semantic similarity between two labels like what REXMP does, it still makes better recommendations than REXMP does. The reason is that in KDGAN, $T$ provides $C$ with soft labels at training. The soft labels contain a rich similarity structure over tags which cannot be modeled well by any pairwise similarity between tags used in REXMP. For example, an image labeled with a tag `volleyball` is supplied with a soft label assigning a probability of $10^{-2}$ to `basketball`, $10^{-4}$ to `baseball`, and $10^{-8}$ to `dragonfly`. The reason that $T$ generalizes is reflected in the relative probabilities over tags, which can be used for guiding $C$ to generalize better.

Next, we compare the training curves of NaGAN, KDGAN-WO-GM, and KDGAN. We only plot the performance measured by P@3 in Figure 3b because the other measures exhibit similar training curves. We find that KDGAN learns a more accurate classifier with a smaller number of training epochs (about 100 epochs) than NaGAN (about 220 epochs) and KDGAN-WO-GM (about 150 epochs). After convergence, KDGAN consistently outperforms the best baseline REXMP.

Last, we investigate how the performance of KDGAN varies against the hyperparameters over the YFCC100M dataset. The results are summarized in Figure 5, which are consistent with our observations in the task of deep model compression.

# 5   Conclusion

We proposed a framework named KDGAN to distill knowledge with generative adversarial networks for multi-label learning with privileged provision. We have defined the KDGAN framework as a minimax game where a classifier, a teacher, and a discriminator are trained adversarially. We have proved that the minimax game has an equilibrium where the classifier perfectly models the true data distribution. We use the concrete distribution to control the variance of gradients during the adversarial training and obtained low-variance gradient estimates to accelerate the training. We have shown that KDGAN outperforms the state-of-the-art methods in two important applications, image tag recommendation and deep model compression. We show that KDGAN learns a more accurate classifier at a faster speed than a naive GAN (NaGAN) does. For future work, we will explore adaptive methods for determining model hyperparameters to achieve better training dynamics.

## Acknowledgement

This work is supported by Australian Research Council Future Fellowship Project FT120100832 and Discovery Project DP180102050. We thank the anonymous reviewers for their feedback on the paper. We have incorporated responses to reviewers' comments in the paper.

## Footnotes

[2]  The Gumbel distribution can be sampled by drawing $\boldsymbol{u} \sim \mathrm{Uniform}(0, 1)$ and computing $\boldsymbol{g} = -\log(-\log \boldsymbol{u})$.

[3]  The code and the data are made available at `https://github.com/xiaojiew1/KDGAN/`.

[4] `https://www.flickr.com/`.    [5] Yahoo Webscope Program. `http://webscope.sandbox.yahoo.com/`.

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
