[Supplementary Material]

# A  Theoretical Analysis

In this section, we provide detailed proofs of the theoretical results in the KDGAN framework. Let $p_\alpha^\varrho(\boldsymbol{y}|\boldsymbol{x}) = \alpha p_c(\boldsymbol{y}|\boldsymbol{x}) + (1 - \alpha)p_t^\varrho(\boldsymbol{y}|\boldsymbol{x})$, which is referred to as the mixture distribution. We first show that the optimal distribution of the discriminator balances between the true data distribution $p_u(\boldsymbol{y}|\boldsymbol{x})$ and the mixture distribution $p_\alpha^\varrho(\boldsymbol{y}|\boldsymbol{x})$, as stated below.

**Lemma 4.1.** *For any fixed classifier and teacher, the value function $U(c, t, d)$ is maximized if and only if the distribution of the discriminator is given by $p_d^\varrho(\boldsymbol{x}, \boldsymbol{y}) = {p_u(\boldsymbol{y}|\boldsymbol{x})}/{(p_u(\boldsymbol{y}|\boldsymbol{x}) + p_\alpha^\varrho(\boldsymbol{y}|\boldsymbol{x}))}$.*

*Proof.* Given the classifier $p_c(\boldsymbol{y}|\boldsymbol{x})$ and the teacher $p_t^\varrho(\boldsymbol{y}|\boldsymbol{x})$, the discriminator aims to maximize the value function $U(c, t, d)$ of the minimax game as

$$\max_d U(c, t, d)$$

$$= \mathbb{E}_{\boldsymbol{y} \sim p_u}[\log p_d^\varrho(\boldsymbol{x}, \boldsymbol{y})] + \alpha \mathbb{E}_{\boldsymbol{y} \sim p_c}[\log(1 - p_d^\varrho(\boldsymbol{x}, \boldsymbol{y}))] + (1 - \alpha)\mathbb{E}_{\boldsymbol{y} \sim p_t^\varrho}[\log(1 - p_d^\varrho(\boldsymbol{x}, \boldsymbol{y}))]$$

$$= \mathbb{E}_{\boldsymbol{y} \sim p_u}[\log p_d^\varrho(\boldsymbol{x}, \boldsymbol{y})] + \alpha \sum_{\boldsymbol{y}} p_c(\boldsymbol{y}|\boldsymbol{x}) \log(1 - p_d^\varrho(\boldsymbol{x}, \boldsymbol{y})) + (1 - \alpha) \sum_{\boldsymbol{y}} p_t^\varrho(\boldsymbol{y}|\boldsymbol{x}) \log(1 - p_d^\varrho(\boldsymbol{x}, \boldsymbol{y}))$$

$$= \mathbb{E}_{\boldsymbol{y} \sim p_u}[\log p_d^\varrho(\boldsymbol{x}, \boldsymbol{y})] + \sum_{\boldsymbol{y}} \left(\alpha p_c(\boldsymbol{y}|\boldsymbol{x}) + (1 - \alpha)p_t^\varrho(\boldsymbol{y}|\boldsymbol{x})\right) \log(1 - p_d^\varrho(\boldsymbol{x}, \boldsymbol{y}))$$

$$= \mathbb{E}_{\boldsymbol{y} \sim p_u}[\log p_d^\varrho(\boldsymbol{x}, \boldsymbol{y})] + \sum_{\boldsymbol{y}} p_\alpha^\varrho(\boldsymbol{y}|\boldsymbol{x}) \log(1 - p_d^\varrho(\boldsymbol{x}, \boldsymbol{y}))$$

$$= \sum_{\boldsymbol{y}} p_u(\boldsymbol{y}|\boldsymbol{x}) \log p_d^\varrho(\boldsymbol{x}, \boldsymbol{y}) + \sum_{\boldsymbol{y}} p_\alpha^\varrho(\boldsymbol{y}|\boldsymbol{x}) \log(1 - p_d^\varrho(\boldsymbol{x}, \boldsymbol{y}))$$

$$= F(p_d^\varrho(\boldsymbol{x}, \boldsymbol{y})).$$

The function $F(p_d^\varrho(\boldsymbol{x}, \boldsymbol{y}))$ achieves the maximum if and only if the distribution of the discriminator is equivalent to $p_d^\varrho(\boldsymbol{x}, \boldsymbol{y}) = {p_u(\boldsymbol{y}|\boldsymbol{x})}/{p_u(\boldsymbol{y}|\boldsymbol{x}) + p_\alpha^\varrho(\boldsymbol{y}|\boldsymbol{x})}$, completing the proof. $\square$

Next, we show that the equilibrium of the minimax game is achieved if and only if both the classifier and the teacher perfectly model the true data distribution, which is summarized as follows.

**Theorem 4.2.** *The equilibrium of the minimax game $\min_{c,t} \max_d U(c, t, d)$ is achieved if and only if $p_c(\boldsymbol{y}|\boldsymbol{x}) = p_t^\varrho(\boldsymbol{y}|\boldsymbol{x}) = p_u(\boldsymbol{y}|\boldsymbol{x})$. At that point, $U(c, t, d)$ reaches the value $-\log(4)$.*

*Proof.* Let $\mathcal{L}_{\mathrm{MD}} = \beta \mathcal{L}_{\mathrm{DS}}^c(p_c(\boldsymbol{y}|\boldsymbol{x}), p_t^\varrho(\boldsymbol{y}|\boldsymbol{x})) + \gamma \mathcal{L}_{\mathrm{DS}}^t(p_t^\varrho(\boldsymbol{y}|\boldsymbol{x}), p_c(\boldsymbol{y}|\boldsymbol{x}))$. Given the optimal distribution of the discriminator in Lemma 4.1, the classifier and the teacher aim to minimize the value function $U(c, t, d)$ of the minimax game as follows,

$$\min_{s,t} U(c, t, d)$$

$$= \sum_{\boldsymbol{y}} p_u(\boldsymbol{y}|\boldsymbol{x}) \log \frac{p_u(\boldsymbol{y}|\boldsymbol{x})}{p_u(\boldsymbol{y}|\boldsymbol{x}) + p_\alpha^\varrho(\boldsymbol{y}|\boldsymbol{x})} + \sum_{\boldsymbol{y}} p_\alpha^\varrho(\boldsymbol{y}|\boldsymbol{x}) \log(1 - \frac{p_u(\boldsymbol{y}|\boldsymbol{x})}{p_u(\boldsymbol{y}|\boldsymbol{x}) + p_\alpha^\varrho(\boldsymbol{y}|\boldsymbol{x})}) + \mathcal{L}_{\mathrm{MD}}$$

$$= \sum_{\boldsymbol{y}} p_u(\boldsymbol{y}|\boldsymbol{x}) \log \frac{p_u(\boldsymbol{y}|\boldsymbol{x})}{p_u(\boldsymbol{y}|\boldsymbol{x}) + p_\alpha^\varrho(\boldsymbol{y}|\boldsymbol{x})} + \sum_{\boldsymbol{y}} p_\alpha^\varrho(\boldsymbol{y}|\boldsymbol{x}) \log \frac{p_\alpha^\varrho(\boldsymbol{y}|\boldsymbol{x})}{p_u(\boldsymbol{y}|\boldsymbol{x}) + p_\alpha^\varrho(\boldsymbol{y}|\boldsymbol{x})} + \mathcal{L}_{\mathrm{MD}}$$

$$= -\log(4) + \mathcal{L}_{\mathrm{KL}}(p_u(\boldsymbol{y}|\boldsymbol{x})||\frac{p_u(\boldsymbol{y}|\boldsymbol{x}) + p_\alpha^\varrho(\boldsymbol{y}|\boldsymbol{x})}{2}) + \mathcal{L}_{\mathrm{KL}}(p_\alpha^\varrho(\boldsymbol{y}|\boldsymbol{x})||\frac{p_u(\boldsymbol{y}|\boldsymbol{x}) + p_\alpha^\varrho(\boldsymbol{y}|\boldsymbol{x})}{2}) + \mathcal{L}_{\mathrm{MD}}$$

$$= -\log(4) + 2\mathcal{L}_{\mathrm{JS}}(p_u(\boldsymbol{y}|\boldsymbol{x})||p_\alpha^\varrho(\boldsymbol{y}|\boldsymbol{x})) + \beta \mathcal{L}_{\mathrm{DS}}^c(p_c(\boldsymbol{y}|\boldsymbol{x}), p_t^\varrho(\boldsymbol{y}|\boldsymbol{x})) + \gamma \mathcal{L}_{\mathrm{DS}}^t(p_t^\varrho(\boldsymbol{y}|\boldsymbol{x}), p_c(\boldsymbol{y}|\boldsymbol{x})).$$

Here, $\mathcal{L}_{\mathrm{KL}}$ is the Kullback–Leibler divergence. $\mathcal{L}_{\mathrm{JS}}$ is the Jensen-Shannon divergence which is non-negative and reaches zero if and only if $p_u(\boldsymbol{y}|\boldsymbol{x}) = p_\alpha^\varrho(\boldsymbol{y}|\boldsymbol{x})$. The distillation losses $\mathcal{L}_{\mathrm{DS}}^c$ and $\mathcal{L}_{\mathrm{DS}}^t$ such as the L2 loss on logits [7] and the Kullback–Leibler divergence on distributions [23] achieve the minimum at zero if and only if $p_c(\boldsymbol{y}|\boldsymbol{x}) = p_t^\varrho(\boldsymbol{y}|\boldsymbol{x})$. Therefore, the value function $U(c, t, d)$ reaches the minimum at $-\log(4)$ if and only if $p_c(\boldsymbol{y}|\boldsymbol{x}) = p_t^\varrho(\boldsymbol{y}|\boldsymbol{x}) = p_\alpha^\varrho(\boldsymbol{y}|\boldsymbol{x}) = p_u(\boldsymbol{y}|\boldsymbol{x})$, which completes the proof. $\square$

Further, we show that the high variance of a random variance can be reduced with a low-variance random variance, which is summarized in Lemma 4.3.

**Lemma 4.3.** *Let $X$ and $Y$ be random variables with $\mathrm{Var}(X) \leq \mathrm{Var}(Y)$. Let $Z = \lambda X + (1 - \lambda)Y$, then we have $\mathrm{Var}(Z) \leq \mathrm{Var}(Y)$ for all $\lambda \in (0, 1)$.*

*Proof.* Given $\text{Var}(X) \leq \text{Var}(Y)$, the covariance $\text{Cov}(X, Y)$ is less than or equal to $\text{Var}(Y)$ because

$$\text{Cov}(X, Y) \leq |\text{Cov}(X, Y)| \leq \sqrt{\text{Var}(X)\,\text{Var}(Y)} \leq \sqrt{\text{Var}(Y)\,\text{Var}(Y)} \leq \text{Var}(Y).$$

According to the properties of the variance, for all $\lambda \in (0, 1)$, we have

$$\begin{aligned}
\text{Var}(Z) &= \lambda^2 \text{Var}(X) + 2\lambda(1-\lambda)\text{Cov}(X, Y) + (1-\lambda)^2 \text{Var}(Y) \\
&\leq \lambda^2 \text{Var}(Y) + 2\lambda(1-\lambda)\text{Cov}(X, Y) + (1-\lambda)^2 \text{Var}(Y) \\
&\leq \lambda^2 \text{Var}(Y) + 2\lambda(1-\lambda)\text{Var}(Y) + (1-\lambda)^2 \text{Var}(Y) \\
&= \text{Var}(Y),
\end{aligned}$$

This completes the proof. $\qquad\square$

# B  Gradient Derivation

We provide detailed derivations of the gradient computation in the KDGAN framework. Similar to the definition of the concrete distribution $q_c(\boldsymbol{y}|\boldsymbol{x})$ for the classifier in Equation 9, we first define a concrete distribution $q_t^{\varrho}(\boldsymbol{y}|\boldsymbol{x})$ for the teacher as follows,

$$q_t^{\varrho}(\boldsymbol{y}|\boldsymbol{x}) = \text{softmax}(\frac{\log p_t^{\varrho}(\boldsymbol{y}|\boldsymbol{x}) + \boldsymbol{g}}{\tau}), \quad \boldsymbol{g} \sim \text{Gumbel}(0, 1),$$

where $\tau \in (0, +\infty)$ is a temperature parameter and $\text{Gumbel}(0, 1)$ is the Gumbel distribution [31]. The classifier and the teacher generate continuous samples from the concrete distributions $q_c(\boldsymbol{y}|\boldsymbol{x})$ and $q_t^{\varrho}(\boldsymbol{y}|\boldsymbol{x})$, respectively, and then discretize the continuous samples into pseudo labels. The discriminator aims to maximize the probability of correctly identifying the true labels as positive and the pseudo labels as negative. The discriminator is trained to maximize the value function $U(c, t, d)$ of the minimax game by ascending along its gradients

$$\begin{aligned}
&\nabla_d U(c, t, d) \\
&= \nabla_d \big(\mathbb{E}_{\boldsymbol{y} \sim p_u}[\log p_d^{\varrho}(\boldsymbol{x}, \boldsymbol{y})] + \alpha \mathbb{E}_{\boldsymbol{y} \sim p_c}[\log(1 - p_d^{\varrho}(\boldsymbol{x}, \boldsymbol{y}))] + (1-\alpha)\mathbb{E}_{\boldsymbol{y} \sim p_t^{\varrho}}[\log(1 - p_d^{\varrho}(\boldsymbol{x}, \boldsymbol{y}))]\big) \\
&\approx \frac{1}{k}\sum_{i=1}^{k}\big(\nabla_d \log p_d^{\varrho}(\boldsymbol{x}, \boldsymbol{y}_i) + \alpha\nabla_d \log(1 - p_d^{\varrho}(\boldsymbol{x}, \boldsymbol{z}_i^c)) + (1-\alpha)\nabla_d \log(1 - p_d^{\varrho}(\boldsymbol{x}, \boldsymbol{z}_i^t))\big).
\end{aligned}$$

Here, $k$ is the number of samples used to estimate the gradients. The true label $\boldsymbol{y}_i$ is sampled from the true data distribution $p_u(\boldsymbol{y}|\boldsymbol{x})$. $\boldsymbol{z}_i^c = \text{onehot}(\text{argmax}\,\boldsymbol{y}_i^c)$ and $\boldsymbol{z}_i^t = \text{onehot}(\text{argmax}\,\boldsymbol{y}_i^t)$ are pseudo labels where $\boldsymbol{y}_i^c \sim q_c(\boldsymbol{y}|\boldsymbol{x})$ and $\boldsymbol{y}_i^t \sim q_t^{\varrho}(\boldsymbol{y}|\boldsymbol{x})$ are continuous samples.

The classifier aims to generate the pseudo labels that resemble the true labels and predict the soft labels produced by the teacher. The classifier is trained to minimize the value function $U(c, t, d)$ of the minimax game by descending along its gradients

$$\begin{aligned}
\nabla_c U(c, t, d) &= \nabla_c\big(\alpha\mathbb{E}_{\boldsymbol{y} \sim p_c}[\log(1 - p_d^{\varrho}(\boldsymbol{x}, \boldsymbol{y}))] + \beta\mathcal{L}_{\text{DS}}^c(p_c(\boldsymbol{y}|\boldsymbol{x}), p_t^{\varrho}(\boldsymbol{y}|\boldsymbol{x}))\big) \\
&= \alpha\nabla_c \sum_{\boldsymbol{y}} p_c(\boldsymbol{y}|\boldsymbol{x})\log(1 - p_d^{\varrho}(\boldsymbol{x}, \boldsymbol{y})) + \beta\nabla_c\mathcal{L}_{\text{DS}}^c(p_c(\boldsymbol{y}|\boldsymbol{x}), p_t^{\varrho}(\boldsymbol{y}|\boldsymbol{x})) \\
&= \alpha\sum_{\boldsymbol{y}} \nabla_c p_c(\boldsymbol{y}|\boldsymbol{x})\log(1 - p_d^{\varrho}(\boldsymbol{x}, \boldsymbol{y})) + \beta\nabla_c\mathcal{L}_{\text{DS}}^c(p_c(\boldsymbol{y}|\boldsymbol{x}), p_t^{\varrho}(\boldsymbol{y}|\boldsymbol{x})) \\
&= \alpha\sum_{\boldsymbol{y}} p_c(\boldsymbol{y}|\boldsymbol{x})\nabla_c \log p_c(\boldsymbol{y}|\boldsymbol{x})\log(1 - p_d^{\varrho}(\boldsymbol{x}, \boldsymbol{y})) + \beta\nabla_c\mathcal{L}_{\text{DS}}^c(p_c(\boldsymbol{y}|\boldsymbol{x}), p_t^{\varrho}(\boldsymbol{y}|\boldsymbol{x})) \\
&= \alpha\mathbb{E}_{\boldsymbol{y} \sim p_c}[\nabla_c \log p_c(\boldsymbol{y}|\boldsymbol{x})\log(1 - p_d^{\varrho}(\boldsymbol{x}, \boldsymbol{y}))] + \beta\nabla_c\mathcal{L}_{\text{DS}}^c(p_c(\boldsymbol{y}|\boldsymbol{x}), p_t^{\varrho}(\boldsymbol{y}|\boldsymbol{x})) \\
&\approx \frac{\alpha}{k}\sum_{i=1}^{k} \nabla_c \log q_c(\boldsymbol{y}_i^c|\boldsymbol{x})\log(1 - p_d^{\varrho}(\boldsymbol{x}, \boldsymbol{z}_i^c)) + \beta\nabla_c\mathcal{L}_{\text{DS}}^c(p_c(\boldsymbol{y}|\boldsymbol{x}), p_t^{\varrho}(\boldsymbol{y}|\boldsymbol{x})),
\end{aligned}$$

where $\boldsymbol{z}_i^c = \text{onehot}(\text{argmax}\,\boldsymbol{y}_i^c)$ is a pseudo label and $\boldsymbol{y}_i^c \sim q_c(\boldsymbol{y}|\boldsymbol{x})$ is a continuous sample. At the training of the classifier, we use a control variate [49], which is defined as

$$b_c = \mathbb{E}_{\boldsymbol{y} \sim p_c(\boldsymbol{y}|\boldsymbol{x})}[\log(1 - p_d^{\varrho}(\boldsymbol{x}, \boldsymbol{y}))] \approx \sum_{i=1}^{k} \log(1 - p_d^{\varrho}(\boldsymbol{x}, \boldsymbol{z}_i^c)),$$

where $\boldsymbol{z}_i^c = \text{onehot}(\text{argmax}\,\boldsymbol{y}_i^c)$ is obtained by discretizing a continuous sample $\boldsymbol{y}_i^c \sim q_c(\boldsymbol{y}|\boldsymbol{x})$. $\nabla_c\mathcal{L}_{\text{DS}}^c$ is the gradients of the distillation loss $\mathcal{L}_{\text{DS}}^c$ w.r.t. the classifier, which can be easily computed

by the back-propagation algorithm. For example, if we use the L2 loss on logits [7] to define the distillation loss $\mathcal{L}_{\text{DS}}^c$ as

$$\mathcal{L}_{\text{DS}}^c(p_c(\boldsymbol{y}|\boldsymbol{x}), p_t^\varrho(\boldsymbol{y}|\boldsymbol{x})) = \frac{1}{2}||\log p_c(\boldsymbol{y}|\boldsymbol{x})) - \log p_t^\varrho(\boldsymbol{y}|\boldsymbol{x})||_2^2,$$

the gradients $\nabla_c \mathcal{L}_{\text{DS}}^c$ are computed by

$$\nabla_c \mathcal{L}_{\text{DS}}^c(p_c(\boldsymbol{y}|\boldsymbol{x}), p_t^\varrho(\boldsymbol{y}|\boldsymbol{x})) = ||\log p_c(\boldsymbol{y}|\boldsymbol{x})) - \log p_t^\varrho(\boldsymbol{y}|\boldsymbol{x})||_2 \nabla_c \log p_c(\boldsymbol{y}|\boldsymbol{x}).$$

Similarly, the gradients to update the teacher are derived as follows,

$$\nabla_t U(c, t, d) = \nabla_t \big((1 - \alpha)\mathbb{E}_{\boldsymbol{y} \sim p_t^\varrho}[\log(1 - p_d^\varrho(\boldsymbol{x}, \boldsymbol{y}))] + \gamma \mathcal{L}_{\text{DS}}^t(p_t^\varrho(\boldsymbol{y}|\boldsymbol{x}), p_c(\boldsymbol{y}|\boldsymbol{x}))\big)$$

$$= (1 - \alpha) \sum\nolimits_{\boldsymbol{y}} \nabla_t p_t^\varrho(\boldsymbol{y}|\boldsymbol{x}) \log(1 - p_d^\varrho(\boldsymbol{x}, \boldsymbol{y})) + \gamma \nabla_t \mathcal{L}_{\text{DS}}^t(p_t^\varrho(\boldsymbol{y}|\boldsymbol{x}), p_c(\boldsymbol{y}|\boldsymbol{x}))$$

$$= (1 - \alpha) \sum\nolimits_{\boldsymbol{y}} \nabla_t p_t^\varrho(\boldsymbol{y}|\boldsymbol{x}) \log(1 - p_d^\varrho(\boldsymbol{x}, \boldsymbol{y})) + \gamma \nabla_t \mathcal{L}_{\text{DS}}^t(p_t^\varrho(\boldsymbol{y}|\boldsymbol{x}), p_c(\boldsymbol{y}|\boldsymbol{x}))$$

$$= (1 - \alpha) \sum\nolimits_{\boldsymbol{y}} p_t^\varrho(\boldsymbol{y}|\boldsymbol{x}) \nabla_t \log p_t^\varrho(\boldsymbol{y}|\boldsymbol{x}) \log(1 - p_d^\varrho(\boldsymbol{x}, \boldsymbol{y})) + \gamma \nabla_t \mathcal{L}_{\text{DS}}^t(p_t^\varrho(\boldsymbol{y}|\boldsymbol{x}), p_c(\boldsymbol{y}|\boldsymbol{x}))$$

$$= (1 - \alpha)\mathbb{E}_{\boldsymbol{y} \sim p_t^\varrho}[\nabla_t \log p_t^\varrho(\boldsymbol{y}|\boldsymbol{x}) \log(1 - p_d^\varrho(\boldsymbol{x}, \boldsymbol{y}))] + \gamma \nabla_t \mathcal{L}_{\text{DS}}^t(p_t^\varrho(\boldsymbol{y}|\boldsymbol{x}), p_c(\boldsymbol{y}|\boldsymbol{x}))$$

$$\approx \frac{1 - \alpha}{k} \sum\nolimits_{i=1}^k \nabla_t \log q_t^\varrho(\boldsymbol{y}_i^t|\boldsymbol{x}) \log(1 - p_d^\varrho(\boldsymbol{x}, \boldsymbol{z}_i^t)) + \gamma \nabla_t \mathcal{L}_{\text{DS}}^t(p_t^\varrho(\boldsymbol{y}|\boldsymbol{x}), p_c(\boldsymbol{y}|\boldsymbol{x})),$$

where $\boldsymbol{z}_i^t = \text{onehot}(\text{argmax}\,\boldsymbol{y}_i^t)$ is a pseudo label and $\boldsymbol{y}_i^t \sim q_t^\varrho(\boldsymbol{y}|\boldsymbol{x})$ is a continuous sample. At the training of the teacher, we also use a control variate [49], which is defined as

$$b_t = \mathbb{E}_{\boldsymbol{y} \sim p_t^\varrho(\boldsymbol{y}|\boldsymbol{x})}[\log(1 - p_d^\varrho(\boldsymbol{x}, \boldsymbol{y}))] \approx \sum\nolimits_{i=1}^k \log(1 - p_d^\varrho(\boldsymbol{x}, \boldsymbol{z}_i^t)),$$

where $\boldsymbol{z}_i^t = \text{onehot}(\text{argmax}\,\boldsymbol{y}_i^t)$ is obtained by discretizing a continuous sample $\boldsymbol{y}_i^t \sim q_t^\varrho(\boldsymbol{y}|\boldsymbol{x})$. $\nabla_t \mathcal{L}_{\text{DS}}^t$ is the gradients of the distillation loss $\mathcal{L}_{\text{DS}}^t$ w.r.t. the teacher. For example, the gradients $\nabla_t \mathcal{L}_{\text{DS}}^t$ are given by

$$\nabla_t \mathcal{L}_{\text{DS}}^t(p_t^\varrho(\boldsymbol{y}|\boldsymbol{x}), p_c(\boldsymbol{y}|\boldsymbol{x})) = ||\log p_t^\varrho(\boldsymbol{y}|\boldsymbol{x}) - \log p_c(\boldsymbol{y}|\boldsymbol{x})||_2 \nabla_t \log p_t^\varrho(\boldsymbol{y}|\boldsymbol{x}),$$

when the distillation loss $\mathcal{L}_{\text{DS}}^t$ is defined as the L2 loss on logits [7],

$$\mathcal{L}_{\text{DS}}^t(p_t^\varrho(\boldsymbol{y}|\boldsymbol{x}), p_c(\boldsymbol{y}|\boldsymbol{x})) = \frac{1}{2}||\log p_t^\varrho(\boldsymbol{y}|\boldsymbol{x}) - \log p_c(\boldsymbol{y}|\boldsymbol{x}))||_2^2.$$

## C   Network Architectures

We describe network architectures which we use to conduct experiments in deep model compression and image tag recommendation tasks. First, we describe the network architectures in deep model compression task on the MNIST dataset. We implement the scoring function $h(\boldsymbol{x}, \boldsymbol{y})$ as an MLP [27]. The architecture of the MLP is given by

1. An input layer of a 28×28 grayscale image.
2. A stack of 2 fully connected layers with 800 neurons.
3. A softmax layer with 10 classes.

We implement the scoring function $s(\boldsymbol{x}, \boldsymbol{y})$ as a LeNet [27]. The architecture of the LeNet is given by

1. An input layer of a 28×28 grayscale image.
2. A convolutional layer with 32 kernels of size 5×5 and stride 1.
3. A max pooling layer with size 2×2 and stride 2.
4. A convolutional layer with 64 kernels of size5×5 and stride 1.
5. A max pooling layer with size 2×2 and stride 2.
6. A fully connected layer with 1024 neurons.
7. A softmax layer with 10 classes.

Next, we describe the network architectures in deep model compression task on the CIFAR-10 dataset. We implement $h(\boldsymbol{x}, \boldsymbol{y})$ as a LeNet [27]. The architecture of the LeNet is given by

1. An input layer of a 32×32 colored image.
2. A convolutional layer with 64 kernels of size 5×5 and stride 1.
3. A max pooling layer with size 2×2 and stride 2.
4. A convolutional layer with 128 kernels of size 5×5 and stride 1.
5. A max pooling layer with size 2×2 and stride 2.
6. A fully connected layer with 1024 neurons.
7. A softmax layer with 10 classes.

We implement $s(\boldsymbol{x}, \boldsymbol{y})$ as a 101-layer ResNet [22]. The architecture of the ResNet is given by

1. An input layer of a 32×32 colored image.
2. A convolutional layer with 16 kernels with size 3×3 and stride 1.
3. Three stacked blocks of 3 convolutional layers which use 64 kernels of size 1×1, 64 kernels of size 3×3, and 256 kernels of size 1×1, respectively.
4. Four stacked blocks of 3 convolutional layers which use 128 kernels of size 1×1, 128 kernels of size 3×3, and 512 kernels of size 1×1, respectively.
5. Twenty three stacked blocks of 3 convolutional layers which use 256 kernels of size 1×1, 256 kernels of size 3×3, and 1024 kernels of size 1×1, respectively.
6. Three stacked blocks of 3 convolutional layers which use 512 kernels of size 1×1, 512 kernels of size 3×3, and 2048 kernels of size 1×1, respectively.
7. A global pooling layer.
8. A softmax layer with 10 classes.

Finally, we describe the network architectures in image tag recommendation task on the YFCC100M dataset. We use the same network architectures when experimenting with the two datasets of images labeled with the 200 most popular tags and 200 randomly sampled tags, respectively. We implement a VGGNet [40] to extract image features. The architecture of the VGGNet is written as

1. An input layer of a 224×224 colored image.
2. A stack of 2 convolutional layers with 64 kernels of size 3×3 and stride 1.
3. A max pooling layer with size 2×2 and stride 2.
4. A stack of 2 convolutional layers with 128 kernels of size 3×3 and stride 1.
5. A max pooling layer with size 2×2 and stride 2.
6. A stack of 2 convolutional layers with 256 kernels of size 3×3 and stride 1.
7. A max pooling layer with size 2×2 and stride 2.
8. A stack of 2 convolutional layers with 512 kernels of size 3×3 and stride 1.
9. A max pooling layer with size 2×2 and stride 2.
10. A stack of 2 convolutional layers with 512 kernels of size 3×3 and stride 1.
11. A max pooling layer with size 2×2 and stride 2.
12. A fully connected layer with 4096 neurons.
13. A fully connected layer with 4096 neurons.
14. A fully connected layer with 100 neurons.

We implement a LSTM [24] to learn text features. The architecture of the LSTM is written as

$$\boldsymbol{f}_t = \text{sigmoid}(\mathbf{W}_f \cdot [\boldsymbol{h}_{t-1}, \boldsymbol{x}_t] + \boldsymbol{b}_f),$$
$$\boldsymbol{i}_t = \text{sigmoid}(\mathbf{W}_i \cdot [\boldsymbol{h}_{t-1}, \boldsymbol{x}_t] + \boldsymbol{b}_i),$$
$$\boldsymbol{o}_t = \text{sigmoid}(\mathbf{W}_o \cdot [\boldsymbol{h}_{t-1}, \boldsymbol{x}_t] + \boldsymbol{b}_o),$$
$$\boldsymbol{s}_t = \boldsymbol{f}_t \odot \boldsymbol{s}_{t-1} + \boldsymbol{i}_t \odot \tanh(\mathbf{W}_s \cdot [\boldsymbol{h}_{t-1}, \boldsymbol{x}_t] + \boldsymbol{b}_s),$$
$$\boldsymbol{h}_t = \boldsymbol{o}_t \odot \tanh(\boldsymbol{s}_t),$$

where $[\boldsymbol{h}, \boldsymbol{x}]$ is the vector concatenation and $\odot$ is the element-wise product. We set the hidden size of the LSTM to 100 in the experiments. Let $\boldsymbol{v}_x \in \mathbb{R}^{100}$ be an image feature vector extracted by the VGGNet and $\boldsymbol{v}_z \in \mathbb{R}^{100}$ be a text feature vector learned by the LSTM. We implement the scoring function $h(\boldsymbol{x}, \boldsymbol{y})$ as an MLP [3]. The architecture of the MLP is written as

1. An input layer of a feature vector with size 100 (i.e. the image features $\boldsymbol{v}_x$).
2. A stack of 2 fully connected layers with 800 neurons.
3. A softmax layer with 200 classes.

We implement the scoring function $s(\boldsymbol{x}, \boldsymbol{y})$ as an MLP [3]. The architecture of the MLP is given by

1. An input layer of a feature vector with size 100 (i.e. the element-wise product of $\boldsymbol{v}_x$ and $\boldsymbol{v}_z$).
2. A stack of 2 fully connected layers with 1200 neurons.
3. A softmax layer with 200 classes.

# D    Additional Experiments

(a) Using 100 training images.

(b) Using 10,000 training images.

Figure 6: The accuracy of the teacher against the hyperparameter $\gamma$ in KDGAN on MNIST. Note that $\gamma$ controls how much the classifier distills its knowledge into the teacher.

(a) KDGAN without the Gumbel-Max trick.

(b) KDGAN with Gumbel-Max trick.

Figure 7: Variances of the gradient of the adversarial loss ($\nabla_c \mathcal{L}_{\mathrm{AD}}^n$) or the distillation loss ($\nabla_c \mathcal{L}_{\mathrm{DS}}^c$) w.r.t. the classifier. The results are obtained by training KDGAN with 100 training images on MNIST.

We study the classifier's ratio of compression (in terms of the number of parameters) and loss of accuracy w.r.t. the teacher. The results using 5K to 50K training images on MNIST are presented in Tables 3 and 4. We can see that the loss of accuracy generally decreases as the parameter number of the classifier or the number of training examples increases. We also observe that MIMIC (1.22M parameters) achieves an accuracy of 97.93%-99.05% while KDGAN with a much smaller size (0.19M parameters) already achieves a better accuracy (98.72%-99.27%) than MIMIC.

Table 3: Model size and accuracy of the classifier and the teacher (shown in parenthesis) in KDGAN on MNIST.

| #Param. (M) | $n = 5$K | $n = 10$K | $n = 50$K |
|---|---|---|---|
| 0.09 (3.12) | 97.96 (99.25) | 98.74 (99.42) | 99.03 (99.65) |
| 0.19 (3.12) | 98.72 (99.26) | 98.92 (99.46) | 99.27 (99.70) |
| 1.22 (3.12) | 99.01 (99.28) | 99.25 (99.48) | 99.54 (99.72) |
| 2.28 (3.12) | 99.04 (99.27) | 99.40 (99.53) | **99.77 (99.78)** |

Table 4: Average accuracy over 10 runs with varying training size ($n$) on MNIST.

| Method | $n = 5$K | $n = 10$K | $n = 50$K |
|---|---|---|---|
| CODIS | 98.53 | 98.89 | 99.31 |
| DISTN | 98.04 | 98.79 | 99.26 |
| MIMIC | 97.93 | 98.65 | 99.05 |
| KDGAN | **99.01** | **99.25** | **99.54** |