[Reviews · NeurIPS 2018]

Reviewer 1



In this paper, the authors propose combining a knowledge distillation and GANs to improve the accuracy for multi-class classification. (In particular, the GANs are most related to IRGAN, where the discriminator is distinguishing between discrete distributions rather than continuous ones.) At the core, they demonstrate that combining these two approaches provides a better balance of sample efficiency and convergence to the ground truth distribution for improved accuracy. They claim two primary technical innovations (beyond combining these two approaches): using the Gumbel-Max trick for differentiability and having the classifier supervise the teacher (not just the teacher supervise the classifier). They argue that the improvements come from lower variance gradients and that the equilibrium of the minimax game is convergence to the true label distribution. The idea of combining these two perspectives is interesting, and both the theoretical arguments and the empirical results are compelling. Overall, I like the paper and think others will benefit from reading it. However, I think there are a few gaps in the analysis that if closed would greatly benefit the paper: (1) Because there are a multiple design decisions made that are claimed to balance for an improvement, it is important to show that it is not one particular change that is causing the majority of the benefit. In particular, using the classifier to jointly train teacher has been published in "Large scale distributed neural network training through online distillation" (which should be cited but is not). Comparing KDGAN with co-distillation would be an interesting baseline (this is missed in the hyperparameter analysis because of the way alpha parameterizes the GAN loss). I think this is important as co-distillation even without GAN will allow for the true label distribution to be a convergence point, while maintaining low-variance updates. (1a) While the hyperparameter sweeps provide a useful analysis of where the benefits come from, another (though in my opinion less interesting) missing component is understanding how much benefit the Gumbel-Max trick provides. (2) Lower variance of KD than GAN: Section 3.3 claims that the gradients from the KD loss are lower variance than the gradients from the GAN loss. While this is intuitively believable, and prior work in this area is cited as support, I don't see any proof or empirical evidence adding to this claim. There are some slight notation improvements that could be made, such as being consistent as to whether D references the discriminator or the distillation (see line 195). However, overall the paper is clear and the results are valuable.

Reviewer 2



The papers performs some knowledge distillation between two networks using GAN related loss. The trick is to consider the output from a classifier as a generative process fed by the examples to classify. Then using a discriminator it becomes possible to train the "student" network to be undisguisable from the generative process from the "teacher". pro: - Related work section is nicely done and very pleasant to read - The paper uses some recently introduced tricks as the Gumbel-max to tackle the difficulty to learn GAN with discrete distributions - Thanks to soft label annealing it seems easier to learn the student network. This point could deserve more attention since an obvious discriminator is the scalar product of the one hot encoded classes emitted by the two generators cons: - The papers claims to be able to learn lighter classifiers but never discusses classical elements of KD as compression or loss of precision. - Is the paper aiming to reduce the required number of samples to achieve learn a classifier? If YES it would be good to compare to works close to one-shot learning (see https://arxiv.org/pdf/1606.04080.pdf as an example which have higher accuracy on harder datasets with very few examples). If NOT, the discussion about the number of samples becomes somewhat irrelevant unless transformed into something like "number of samples requested to exhibit KD" and then prove empirically that there is scenarios where the proposed approach requires less samples. - I'm not sure to understand why the initial input x should be provided to the discriminator - The reported scores for the final student nets are quite low both on MNIST and CIFAR. Thought I understand this is not the sole objective of this work, I feel this cast a shadow on the the empirical evidences of this paper.

Reviewer 3



The authors propose a novel framework composed of three players, a classifier, a teacher, and discriminator. With this work, this paper effectively combines distillation and adversarial networks in one frame. The classifier is not allowed to privileged provision (such as larger capacity model) but train from the teacher with distillation loss. Both teacher and classifier is trained on adversarial loss against the discriminator, following standard GAN setting. The authors proposed KDGAN and showed it is effective and efficient in two interesting problems, model compression and image tag recommendation task. Strength: - This paper is well-written in general. It summarizes existing work (especailly Naive GAN) clearly, so the readers can easily understand the background. KDGAN formulation was also clearly motivated and explained. - It is especially good to have a section 3.3, trying to improve training speed with theoretical motivation. - Both experiments are relevant and important applications in ML field. - I believe this paper is worth to be presented at NIPS in terms of originality and clarity. Weakness/Suggestion: - In section 3.2, the authors mentioned that the teacher model also learns from classifier (student) model, motivating from a general thought that teachers can also learn from their students in society. It'd be nicer to see some experimental evidence to show this. - In section 4.1, the experiment is about deep model compression, aiming at reducing the storage and runtime complexity. I see experiments about faster training, while I couldn't see discussion about the size of the model. It will be interesting to compare smallest model sizes accomplishing similar level of accuracy, though this experiment may be harder to conduct. - Image tag recommendation may be easily applied to video annotation dataset (e.g, YouTube 8M) for your future work. [Additional comment after author response] Thanks for additional experiment and clear explanations. I adjusted the score, as it cleared my concerns.

Reviewer 4



This paper describes a method to integrate knowledge distillation and adversarial network to get the best of both worlds. The proposed architecture has three components: a classifier, a teacher and a discriminator. A distillation loss regulates the relation between the classifier and the teacher so that the produced pseudo-labels agree with each other. The adversarial loss connects the teacher and the discriminator: the last one tries to recognize pseudo-labels from true labels. The difference of the proposed strategy with respect to pre-existing methods with three similar components are well discussed in the related work section. Besides that, two novelties are intruduced: (1) mutual learning of T and C (not only C from T but also T from C); (2) the Gumbel-Max trick to reduce the variance of gradients. Overall the paper introduces some new ideas and the experiments show the effectiveness of the proposed strategy. I have only few doubts: - I find it a bit confusing the discussion about the availability of 'priviledged information' at training time as basic motivation of the work. Indeed when passing to the experimental sections we are only dealing with standard samples x and standard labels y, there is no mention of extra information which are generally defined as priviledged (e.g. attributes). - the first set of experiments is about model compression, however details about the obtained compression are not provided (number of parameters of T and C?). It is not fully clear to me if the considered competing methods are actually obtaining the same model compression, or wheather KDGAN is more/less effective in this sense (more compression) besides being better in terms of produced accuracy. - Although the introduction of the two novelties mentioned above surely helps increasing the overall KDGAN performance it is not explicitly discussed their role. An ablation analysis would be beneficial to understand quantitativaly their contribution.